# Content of Polyphenolic Compounds and Antioxidant Potential of Some Bulgarian Red Grape Varieties and Red Wines, Determined by HPLC, UV, and NIR Spectroscopy

**Milena Tzanova *** [iD]**, Stefka Atanassova, Vasil Atanasov and Neli Grozeva**

Faculty of Agriculture, Trakia University, Studentski grad Str., 6000 Stara Zagora, Bulgaria; atanassova@uni-sz.bg (S.A.); vka@mail.bg (V.A.); grozeva@uni-sz.bg (N.G.)
* Correspondence: tzanova_m@yahoo.com; Tel.: +359-42-699-315

**Abstract:** Today, good food criteria also include healthy capacity. So, the wine on our table should not only have good organoleptic qualities, but should be characterized by a high healthy potential. For the first time, extensive research was conducted on commercial red wine grape varieties cultivated in different Bulgarian regions in two consecutive years. Antioxidants, including trans-resveratrol, quercetin, and total phenolic content and antioxidant potential in wine grapes and wines were determined by HPLC, UV, and NIR methods. The results obtained showed similar concentration levels compared to the same varieties, produced in other countries. Trans-resveratrol showed the greatest contribution to the radical scavenging capacity. The factor with largest impact on the content of the tested substances was definitely the variety. Among agro-meteorological condition, temperature amplitude, rain fall, and UV irradiation before ripening had strong influences. Maintaining the balance between the level of synthesized and degraded and captured antioxidants during the wine making process was crucial to preserving the antioxidant properties of the final wine product. NIR spectroscopy showed very good accuracy of determination of trans-resveratrol, quercetin, total phenolic content, and the antioxidant activity of tested grape varieties and red wines. It could be a promising technique in the quantification of their antioxidant parameters.

**Keywords:** trans-resveratrol; quercetin; radical scavenging capacity; red wine grape; red wine; HPLC; UV; NIR

## 1. Introduction

The plants produce de novo compounds with small molecules called phytoalexins (from Greek *phytos*—"plant" and *alekein*—"fend off"). These secondary metabolites are responsible for the plant protection against diverse biotic and abiotic factors, like bacteria, fungi, and UV-irradiation [1]. Produced by vine substance called phenolic compounds, these known bio-antioxidants have a high potential [2]. During the process of vinification, they pass from the solid parts of the grapes into the must. Maceration is a very important stage in red wine production.

Trans-resveratrol (t-RVT) belongs to the stilbene polyphenol subgroup and is potentially responsible for the "French paradox"—the French suffer relatively seldom from coronary heart disease, although their diet is rich in saturated fats [3]. A number of studies, reviewed by Novelle et al., on its influence on the human health as dietary supplement, demonstrated diverse bio-activities like antioxidant, cardio protective, anti-cancer, anti-diabetic, and anti-inflammatory agents [4].

Some selected natural and synthetic compounds are proven to work in synergy with trans-resveratrol. One of them is quercetin (QU, a flavonol, one of the subclasses of flavonoid

compounds). A number of research teams studied and proved its antioxidant potential, as well as its anti-allergy, anticancer, blood pressure lowering, anti-inflammatory and antiviral activities [5]. In addition, quercetin increases the protective effect of trans-resveratrol as an antioxidant [6], an anti-inflammatory [7], a cardiovascular [8] and an anti-obesity agent [9].

The wine industry is currently requiring fast, low cost, and easy-to-operate techniques for routine grape and wine analysis. NIR spectroscopy is a non-destructive method, requiring minimal or no sample preparation, with good abilities in the analysis of different food products. Recently, several review papers can be found in the literature, demonstrating the abilities of NIR spectroscopy for determination of the content of flavanols, anthocyanins, and other phenolic compounds in wine and grapes [10,11].

Bulgarian wines have been on the international market for a long time, but data about the antioxidant properties, phenols, trans-resveratrol and quercetin are deficient. The aim of this study was the determination of their content in some ccommercial Bulgarian red wine grapes and wines by HPLC, UV, and NIR methods. Data about antioxidant constituents and antioxidant potential of a number of red wine grape varieties distributed on the Bulgarian and European market were collected. The impact of various agro-meteorological conditions on their content were established.

## 2. Materials and Methods

### 2.1. Samples

The grapes grown in three different districts in Bulgarian viticulturally regions: Danubian plain in the North-Bulgarian town of Pleven (N: 43.407778° E: 24.620278°, 116 m a.s.l.; moderate continental climate and average growing degree days in the last 3 years—3911 °C) and Thracian lowland in South-Bulgarian Region. Village of Mogilovo, Chirpan municipality (N: 42.333333° E: 25.4°, 312 m a.s.l., moderate continental climate and average growing degree days in the last 3 years—3820 °C), and Village of Mezek, Svilengrad municipality (N: 41.73333° E: 26.08333°, 168 m a.s.l., moderate continental climate and average growing degree days in the last 3 years—4346 °C). The regions provided good soil and climatic conditions for producing red wine grape varieties and wines of above average quality.

From Pleven, eight varieties vintage 2017 were selected: three introduced—Merlot, Cabernet Sauvignon and Syrah, and five hybrids—Rubin, Kaylashki Rubin, Storgozia, Bouquet and Otel. During the vegetation seasons of 2017, no extreme climate events were observed.

From Mogilovo, four introduced grape varieties were tested: Merlot, Cabernet Sauvignon, Syrah, and Malbec—vintage 2017 and 2018 wines, and wine grapes vintage 2018.

From Mezek, three introduced grape varieties were tested: Merlot, Cabernet Sauvignon and Syrah—vintage 2018 wines, and wine grapes vintage 2018.

Guyot training system was applied to all varieties and conventional methods of pest control were used during growing season. When grapes reached technological maturity, they were picked and used to produce red wines, in accordance with the generally accepted wine making procedures, which included the following steps: maceration in stainless tanks for 10–12 days at 7–8 °C; heating up to 23–24 °C, and yeast (*Saccharomyces cerevisiae*) adding; fermentation for 7–8 days at 25 °C–28 °C; post-enzymatic maceration for 10–12 days at 25 °C; wine draining in tanks; malolactic acid fermentation; and finally, wine decanting and maturing in wooden barrels for 10–12 months. Three samples of each batch were taken by the enologist of the corresponding winery at the moment of wine aging and were stored in amber glass bottles with cork stopper in a dark place, at 18 °C for one week prior to the analysis. Every sample was analyzed in triplicate.

Grape samples were collected in the period from the middle of September to the beginning of October in 2018, at the moment of reaching of technological maturity. Small portions of the base, middle, and top of the grape cluster, located at the base, middle, and top of the vine were picked. Grapes free from visible blemish or disease were selected. Samples consisted of 3–5 kg of grape

berries. After sampling, the grapes were peeled. The skins were immediately frozen, transported in a refrigerator, and stored for a maximum of 7 days at −12 °C prior to the analysis.

## 2.2. HPLC Analysis

For the quantification of trans-resveratrol and quercetin in grape berry skins, the method validated by Tzanova and Peeva [12] was applied: a sample of 10 g thawed, at room temperature, berry skins were weighed to the nearest ±0.0001 g, grinded and homogenized with a mechanical homogenizer in 60 mL of a 1% solution of HCl in methanol for 5 min in a dark room. The dispersed samples were left in the dark at room temperature overnight and then filtered through a 0.45 μm membrane. The solid residue was washed twice with 10 mL of 1% HCl in methanol. The extracts were collected and adjusted to 100 mL with solvent, and stored at −12 °C until initiation of the HPLC analysis. After filtration, through a 0.45 μm membrane, a sample of 1 mL of each extract was placed into the HPLC auto sampler, to assure the injection volume of 20 μL.

Five-point calibration graphs (0.05; 0.50; 1.0; 2.0 and 5.0 mg/L) were made in advance using reference materials—trans-resveratrol HPLC grade (not less than 99%) and quercetin HPLC grade (not less than 98%), purchased from Sigma-Aldrich (St. Louis, MO, USA). An excellent linear dependence with regression coefficients ($r^2$) 0.9991 for t-RVT and 0.9999 for QU was achieved. Analytical HPLC was performed with a C18 column Hypersil Gold (5μm; 150 mm × 4.6 mm) on a Thermo system, composed of a Surveyor LC Pump Plus, Surveyor Autosampler Plus, and Surveyor photodiode array detector PDA Plus. Chromatograms were recorded at 289 nm for QU and 306 nm for t-RVT in a 7-min single run, with retention times of approximately 4.7 min for RVT and 5.6 min for QU (Figure 1). Results were calculated as mg t-RVT per kg of starting fresh material (FM).

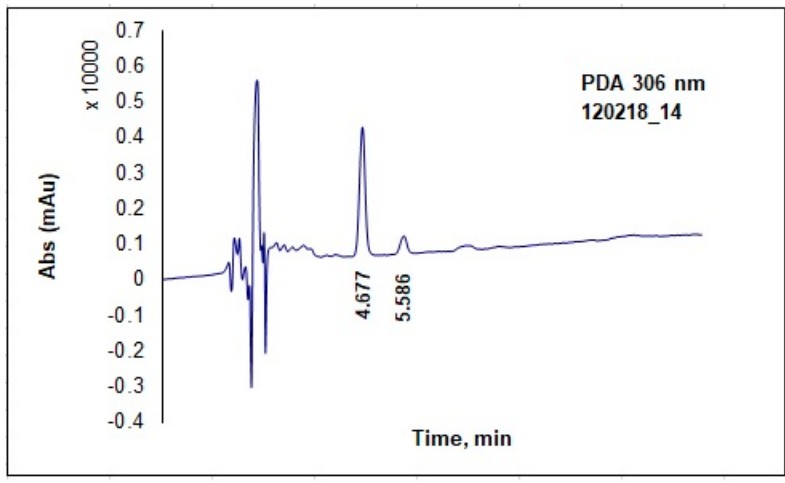

**Figure 1.** Typical chromatogram of sample solution.

## 2.3. UV Analysis

UV methods were applied to determine the total phenol content (TPC) and the antioxidant potential of grape berry skin extract with a final concentration of 1 mg/mL dry matter and of wine direct after filtration through a 0.45 μm membrane by following the procedures described by Tzanova et al. [13]. Defrosted skins were grinded, sample of 5 g was suspended in 50 mL methanol and the target compounds were extracted by ultrasonication for 30 min at 40 °C in triplicate.

Briefly, for the TPC determination: 0.5 mL of the sample solution was mixed in separate tubes with 2.5 mL of Folin-Ciocalteu's reagent (1:10 dilution with water of the commercial reagent). Then, 2 mL of $Na_2CO_3$ in water (7.5% w/v) was added and the tubes were left at room temperature for 60 min. The absorbance at 765 nm was measured against water on a Thermo Scientific Evolution 300 spectrophotometer. Each sample was analyzed in triplicate. Gallic acid (Sigma-Aldrich, St. Louis, MO) solutions in methanol ranging from 2 to 60 mg/L were used for the calibration curve ($r^2$ = 0.9958). TPC

of each sample was expressed as mmol gallic acid equivalents (GAE) in 1 kg for the grape berry skins (FW) and as mmol GAE in 1 l for the wine samples. The results were calculated by regression analysis of the calibration curve. The milligrams of GAE were converted into millimoles by using the molar mass of gallic acid, 170.12 g/mol.

The antioxidant potential of the skin extracts and the wine samples was established by determining the radical scavenging capacity by the DPPH method (1,1'-difenyl-2-picrylhydrazyl radical). Briefly, to 3.9 mL of a 100 μM solution of DPPH in methanol, 0.1 mL of sample solution was added. After 30 min reaction time, the absorption at 517 nm was measured against methanol on a Thermo Scientific Evolution 300 spectrophotometer. The results for the radical scavenging capacity of the sample solutions were compared with that of Trolox (water-soluble analogue of Vitamin E) and calculated by regression analysis from the linear dependence between concentration of Trolox. The results were expressed as mmol of Trolox equivalent (TE) in 1 kg for the grape berry skins (FW), and as mmol TE in 1 l for the wine samples.

## 2.4. NIR Spectral Measurements and Data Analysis

A NIRQuest 512 spectrometer (Ocean Optics, Inc) in the region 900–1700 nm was used for near-infrared spectral analysis. Wine samples were measured using transmittance mode and 10 mm cuvette. The spectra of grape berry skins were obtained using reflectance fiber-optics probe. Five different spectral measurements of each sample were made and averaged.

A software program Pirouette Version 4.5 (Infometrix, Woodinville, WA, USA) was used for spectral data processing and developing regression equations for polyphenolic compounds and antioxidant potential determination by PLS regression. Spectral data of wine and grape berry skins were transformed as a first derivative before PLS regression. The calibration equations for each parameter were developed and validated with leave-one-out cross validation. Leave-one-out method is recommended when a few samples are used to build the calibration equations. One sample from calibration data set was left-out and an equation was built with the remaining samples. This equation was used to predict tested parameter for the omitted sample. This procedure is repeated until each sample was used once as validation sample and correlation between predicted and reference values and standard error of cross-validation SECV was calculated.

The prediction capacity of each calibration equation was evaluated using statistical parameters from calibration procedure; R—multiple correlation coefficients between reference values and NIR predicted values, SEC—standard error of calibration, SECV—standard errors of cross validation, and the ratio performance deviation (RPD) parameter—the relationship between the standard deviation of the chemical method (SD) and the standard error of cross-validation SECV of the NIR equation. The RPD evaluated the prediction errors in light of the standard deviation of the reference data and thus enables a comparison between models for constituents with different variation ranges. RPD between 2.0 and 2.5 indicates good prediction; RPD between 2.5 and 3.0 indicates very good prediction; and RPD >3 indicates excellent prediction.

## 2.5. Statistical Data Analysis

The statistical analysis was performed using Statistica 6 for Windows. All analytical determinations were performed in triplicate and the mean values ± standard deviation (SD) was reported. One–way ANOVA and Tukey post-hoc multi comparison tests were used for data analysis.

## 3. Results

### 3.1. Content of Polyphenolic Compounds and Antioxidant Potential of Red Grape Varieties

In the present study, red wine grape varieties from two different regions with different agro-meteorological conditions were analyzed. The chosen varieties—Cabernet Sauvignon, Merlot, Syrah and Malbec, are worldwide distributed and were introduced and cultivated in Bulgaria.

The results obtained for the tested wine grapes were presented in Table 1.

**Table 1.** Results * of antioxidant parameters of red wine grapes, vintage 2018.

| ID | Sample | | | t-RVT | QU | GAE | TE |
|---|---|---|---|---|---|---|---|
| | Variety | Vineyard | Batch Numbers | mg/kg | mg/kg | mmol/kg | mmol/kg |
| CSM | Cabernet Sauvignon | Mogilovo | 6 | 3.13 ± 0.33 [cd] | 0.63 ± 0.05 [bc] | 47 ± 6 [c] | 23.2 ± 1.7 [bc] |
| SM1 | Syrah | Mogilovo-Dabovets | 4 | 12.73 ± 1.14 [bc] | 2.02 ± 0.15 [bc] | 194 ± 18 [b] | 48.7 ± 5.1 [b] |
| SM2 | Syrah | Mogilovo-Shipka | 3 | 12.58 ± 1.25 [bc] | 1.87 ± 0.17 [a] | 202 ± 19 [b] | 48.5 ± 4.8 [b] |
| McM | Malbec | Mogilovo | 5 | 13.23 ± 1.28 [bc] | 0.74 ± 0.08 [b] | 178 ± 17 [b] | 43.2 ± 4.4 [ab] |
| MtM | Merlot | Mogilovo | 7 | 6.77 ± 0.57 [b] | 0.78 ± 0.08 [b] | 33 ± 4 [cd] | 38.5 ± 3.9 [a] |
| SS | Syrah | Mezek, Svilengrad | 10 | 14.57 ± 1.34 [bc] | 2.19 ± 0.19 [bc] | 182 ± 17 [b] | 47.3 ± 4.1 [b] |
| CSS | Cabernet Sauvignon | Mezek, Svilengrad | 10 | 4.07 ± 0.37 [c] | 0.61 ± 0.05 [bc] | 48 ± 4 [c] | 27.3 ± 1.8 [bc] |
| MtS | Merlot | Mezek, Svilengrad | 10 | 7.33 ± 0.65 [b] | 0.85 ± 0.07 [b] | 38 ± 3 [cd] | 37.8 ± 3.9 [a] |
| | | *P* ANOVA | | ** | ** | ** | ** |

t-RVT—trans-resveratrol; QU—quercetin; GAE—gallic acid equivalent; TE—trolox equivalent. * All data were given as mean ± standard deviation (SD), and analyzed by ANOVA (** $p \leq 0.001$). Within each column, different letters denote significant differences according to Tukey post-hoc multi comparison tests ($p \leq 0.05$).

The attractive compound trans-resveratrol in the skins of red wine grapes ranged from 3.13 ± 0.33 to 14.57 ± 1.34 mg/kg FM. The superior variety was Syrah, from Mezek (14.57 ± 1.34 mg/kg FM), followed by Malbec, (13.23 ± 1.28 mg/kg FW), Syrah, from Mogilovo-Dabovets (12.73 ± 1.14 mg/kg FW) and Syrah, from Mogilovo-Shipka (t-RVT content 12.58 ± 1.25 mg/kg FW).

Quercetin amounts ranged from 0.61 ± 0.05 mg/kg FW in Cabernet Sauvignon from Thracian Lowland, village of Mezek, to 2.19 ± 0.19 mg/kg FW in Syrah from Mezek. The total phenolic content was found to be in the range from 33 ± 4 mmol/kg in Merlot from Mogilovo to 202 ± 19 mmol/kg in Syrah from the same village. The radical scavenging capacity was found to be from 23.2 ± 1.7 mmol/kg in Cabernet Sauvignon from Mogilovo to 48.7 ± 5.1 mmol/kg in Syrah from the same village.

### 3.2. Content of Polyphenolic Compounds and Antioxidant Potential of Red Wines

The red wines tested and the results obtained were presented in Table 2. Wine produced from the wine grape tested in the present study and wines from two consecutive years were analyzed. The impact of the wine making process and the impact of the agro-meteorological conditions were established.

In the vintage 2017 wines, the values of t-RVT were from 1.03 ± 0.08 mg/L (in Otel from Pleven) to 5.95 ± 0.51 mg/L (in Syrah from Mogilovo-Dabovets); and in vintage 2018 wines, from 1.23 ± 0.12 mg/L (in Cabernet Sauvignon from Mezek) to 3.86 ± 0.31 mg/L (in Syrah from Mogilovo-Dabovets).

QU content ranged from 0.84 ± 0.05 mg/L (in Otel from Pleven) to 3.44 ± 0.32 mg/L (in Syrah from Mogilovo-Dabovets); and from 0.63 ± 0.05 mg/L (in Cabernet Sauvignon from Mezek), to 2.24 ± 0.19 mg/L (in Syrah from Mezek) in the vintage 2017 and 2018 wines, respectively.

Total phenolic content was found to be in the range from 1.61 ± 0.14 mmol GAE/L (in Otel from Pleven) to 10.47 ± 1.05 mmol GAE/L (in Malbec from Mogilovo) in vintage 2017 wines; and in the range from 4.15 ± 0.39 mmol GAE/L (in Malbec from Mogilovo) to 5.28 ± 0.50 mmol GAE/L (in Merlot from Mogilovo) in vintage 2018 wines.

The antioxidant potential of the wines vintage 2017 ranged from 0.32 ± 0.03 mmol TE/L (Otel from Pleven) to 0.93 ± 0.09 mmol TE/L (Malbec from Mogilovo). The TE values of the wines vintage 2018 were measured in the range from 0.59 ± 0.05 mmol TE/L (Cabernet Sauvignon from Mezek) to 0.77 ± 0.08 mmol TE/L (Syrah from Mogilovo-Dabovets).

### 3.3. Determination of Content of Polyphenolic Compounds and Antioxidant Potential of Grapes and Wines by NIR Spectroscopy

Statistical parameters from calibration procedures for NIRS determination of tested constituents and antioxidant potential of examined red wine samples and grape berry skins were presented in Table 3.

**Table 2.** Red wines, vintage 2017 and 2018.

| | Sample | | | t-RVT | QU | GAE | TE |
|---|---|---|---|---|---|---|---|
| ID | Variety | Vineyard | Batch Numbers | mg/L | mg/L | mmol/L | mmol/L |
| **Vintage 2017** | | | | | | | |
| **CSMW17** | Cabernet Sauvignon | Mogilovo | 6 | 2.05 ± 0.18 [bc] | 2.43 ± 0.22 [ab] | 8.17 ± 0.76 [ac] | 0.77 ± 0.05 [ab] |
| **SM1W17** | Syrah | Mogilovo-Dabovets | 4 | 5.95 ± 0.51 [cd] | 3.44 ± 0.32 [c] | 9.53 ± 0.85 [cd] | 0.91 ± 0.08 [c] |
| **McMW17** | Malbec | Mogilovo | 5 | 5.22 ± 0.44 [cd] | 2.78 ± 0.26 [bc] | 10.47 ± 1.05 [d] | 0.93 ± 0.09 [c] |
| **MtMW17** | Merlot | Mogilovo | 7 | 4.65 ± 0.38 [bc] | 2.21 ± 0.19 [ab] | 7.55 ± 0.62 [cd] | 0.90 ± 0.07 [c] |
| **CSPW17** | Cabernet Sauvignon | Pleven | 2 | 1.94 ± 0.13 [ab] | 1.41 ± 0.11 [b] | 2.86 ± 0.21 [c] | 0.34 ± 0.04 [bc] |
| **SPW17** | Syrah | Pleven | 2 | 4.66 ± 0.40 [cd] | 2.33 ± 0.20 [b] | 8.22 ± 0.77 [cd] | 0.85 ± 0.08 [bc] |
| **MtPW17** | Merlot | Pleven | 2 | 3.42 ± 0.25 [bc] | 1.51 ± 0.13 [ab] | 6.45 ± 0.59 [cd] | 0.59 ± 0.05 [bc] |
| **RPW17** | Rubin | Pleven | 2 | 2.06 ± 0.16 [bc] | 1.98 ± 0.17 [b] | 3.13 ± 0.27 [b] | 0.37 ± 0.03 [bc] |
| **KRPW17** | Kaylashki Rubin | Pleven | 2 | 4.22 ± 0.39 [c] | 1.09 ± 0.08 [c] | 8.09 ± 0.75 [cd] | 0.82 ± 0.07 [c] |
| **SaPW17** | Storgozia | Pleven | 2 | 2.23 ± 0.20 [ab] | 1.01 ± 0.06 [c] | 3.67 ± 0.30 [b] | 0.41 ± 0.04 [bc] |
| **BPW17** | Bouquet | Pleven | 2 | 2.34 ± 0.21 [ab] | 1.03 ± 0.07 [c] | 4.03 ± 0.35 [b] | 0.47 ± 0.05 [b] |
| **OPW17** | Otel | Pleven | 2 | 1.03 ± 0.08 [cd] | 0.84 ± 0.05 [cd] | 1.61 ± 0.14 [cd] | 0.32 ± 0.03 [bc] |
| **Vintage 2018** | | | | | | | |
| **CSMW18** | Cabernet Sauvignon | Mogilovo | 6 | 1.75 ± 0.15 [bc] | 1.63 ± 0.12 [a] | 4.72 ± 0.41 [a] | 0.59 ± 0.05 [ab] |
| **SM1W18** | Syrah | Mogilovo-Dabovets | 4 | 3.86 ± 0.31 [bc] | 1.23 ± 0.11 [ab] | 4.37 ± 0.46 [ab] | 0.77 ± 0.08 [ab] |
| **SM2W18** | Syrah | Mogilovo-Shipka | 3 | 3.32 ± 0.29 [b] | 2.21 ± 0.22 [b] | 4.28 ± 0.44 [ab] | 0.71 ± 0.08 [b] |
| **McMW18** | Malbec | Mogilovo | 5 | 3.84 ± 0.30 [bc] | 1.97 ± 0.18 [ab] | 4.15 ± 0.39 [ab] | 0.74 ± 0.07 [ab] |
| **MtMW18** | Merlot | Mogilovo | 7 | 2.32 ± 0.21 [ab] | 1.75 ± 0.16 [ab] | 5.28 ± 0.50 [ab] | 0.62 ± 0.06 [a] |
| **SSW18** | Syrah | Mezek, Svilengrad | 10 | 3.53 ± 0.27 [b] | 2.24 ± 0.19 [b] | 4.55 ± 0.41 [a] | 0.73 ± 0.06 [a] |
| **CSSW18** | Cabernet Sauvignon | Mezek, Svilengrad | 10 | 1.23 ± 0.12 [bc] | 0.63 ± 0.05 [ac] | 4.94 ± 0.42 [a] | 0.51 ± 0.04 [ac] |
| **MtSW18** | Merlot | Mezek, Svilengrad | 10 | 1.94 ± 0.17 [bc] | 1.26 ± 0.09 [ab] | 4.67 ± 0.42 [a] | 0.61 ± 0.05 [a] |
| | *P* ANOVA | | | ** | ** | ** | ** |

All data were given as mean ± standard deviation (SD), and analyzed by ANOVA (** $p \leq 0.001$). Within each column different letters denote significant differences according to Tukey post-hoc multi comparison tests ($p \leq 0.05$). t-RVT—trans-resveratrol; QU—quercetin; GAE—gallic acid equivalent; TE—trolox equivalent.

**Table 3.** Statistical data of the calibration equations for NIR spectroscopy determination of antioxidant constituents and antioxidant activities in red wine grapes and wines.

| Product | Constituent | SEC | Rcal | SECV | Rcv | RPD |
|---|---|---|---|---|---|---|
| **Grapes** | **t-RVT** | 0.447 | 0.993 | 0.424 | 0.994 | 8.49 |
| | **QU** | 0.124 | 0.973 | 0.008 | 0.977 | 4.40 |
| | **GAE** | 12.825 | 0.985 | 12.15 | 0.987 | 5.84 |
| | **TE** | 1.279 | 0.988 | 1.187 | 0.991 | 6.77 |
| **Wines** | **t-RVT** | 0.124 | 0.996 | 0.113 | 0.997 | 10.96 |
| | **QU** | 0.078 | 0.994 | 0.073 | 0.995 | 9.00 |
| | **GAE** | 0.168 | 0.997 | 0.144 | 0.998 | 13.87 |
| | **TE** | 0.014 | 0.997 | 0.015 | 0.998 | 13.71 |

R—multiple correlation coefficients. SEC—standard error of calibration. SECV—standard errors of cross validation. RPD—relationship between the standard deviation of the reference method and the standard error of cross-validation SECV of the NIR equation.

Graphical illustrations of accuracy of determination of t-RVT in wine and grape berry skins are presented in Figure 2.

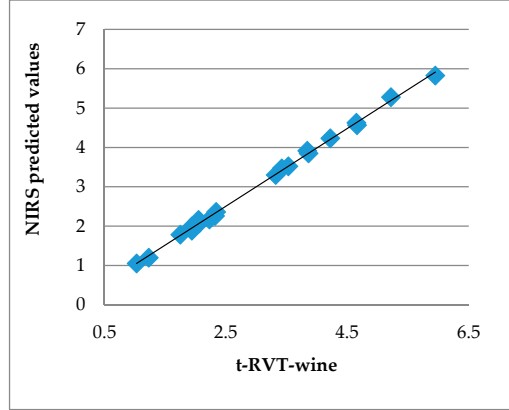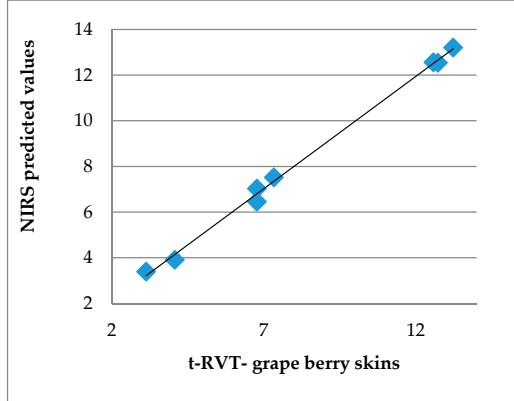

**Figure 2.** Scatter plots of HPLC measured and NIR spectroscopy predicted values of t-RVT in red wine and grape berry skins.

## 4. Discussion

*4.1. Content of Polyphenolic Compounds and Antioxidant Potential of Red Grape Varieties and Corresponding Wines*

The main biotic and abiotic factors influencing the synthesis and accumulation of t-RVT were already specified: varieties and variety clones, agro-meteorological condition, and fungal infections [14]. Great attention was devoted worldwide to the varieties Syrah, Merlot, and Cabernet Sauvignon. The results reported were ambivalent. Rockenbach et al. [15] reported higher t-RVT and QU content in Merlot grape pomace, but higher TPC and greater radical scavenging activity of Cabernet Sauvignon grape pomace varieties "widely produced in Brazil". Zhu et al. [16] found higher stilbene content in Merlot grape skins than in Cabernet Sauvignon from East Asian and North American regions, but the second variety was characterized by larger TPC in its grape skins. Iacopini et al. [17] tested seven grape varieties from Italia and reported 2-fold higher t-RVT content in Cabernet Sauvignon than in Merlot grape skins and similar QU and TPC content values. The results obtained in the present study confirm this statement: for one and the same region t-RVT amount degreased in the following order Syrah, Merlot, and Cabernet Sauvignon. The QU were similar and the TPC assayed, as mmol GAE/kg differed greatly in the same wine region.

One statement more was confirmed: the accumulation of trans-resveratrol in the skin of red grapes is higher than that of quercetin [12,18]. QU content in the grape skins ranged from 0.61 ± 0.05 to 2.19 ± 0.19 mg/kg FW (Table 1).

Interesting results were obtained regarding total phenolic content. TPC, expressed as mmol GAE/kg, ranged within wide limits from one and the same vineyard: from 33 ± 4 to 202 ± 19 and from 38 ± 3 to 182 ± 17 (Table 1). According to TPC, Syrah was the favorite variety and Merlot had less than Cabernet Sauvignon. Ma et al. [18] obtained similar trend; Cabernet Sauvignon variety was richer in phenolic compounds than Merlot.

The antioxidant potential established as radical scavenging capacity ranged in the same region from 23.2 ± 1.7 to 48.7 ± 5.1 mmol TE/kg and from 27.3 ± 1.8 to 47.3 ± 4.1 mmol TE/kg (Table 1). Syrah was superior.

In the present study, the correlation between trans-resveratrol and quercetin content was found to be positive, as well as their relationship to the antioxidant potential, measured by DPPH method. The correlation coefficients were found to be very high between t-RVT, QU and TPC and the radical scavenging activity (Table 4). The coefficient of correlation between TE and t-RVT was found to be the highest, 0.9384, so the contribution of trans-resveratrol to the antioxidant potential was greater than this of QU (0.7959) and TPC (0.8142).

**Table 4.** Correlation matrix of antioxidant parameters of red wine grapes and corresponding wines, vintage 2018.

| Parameters | | Wine Grape, Vintage 2018 | | | | Wine, Vintage 2018 | | | |
|---|---|---|---|---|---|---|---|---|---|
| | | **t-RVT** | **QU** | **GAE** | **TE** | **t-RVT** | **QU** | **GAE** | **TE** |
| **Wine** | **t-RVT** | 1.0000 | 0.7783 | 0.9160 | 0.9384 | 0.9480 | | - | - |
| **grape,** | **QU** | | 1.0000 | 0.7891 | 0.7959 | - | 0.4452 | - | - |
| **vintage** | **GAE** | | | 1.0000 | 0.8141 | - | - | −0.8285 | - |
| **2018** | **TE** | | | | 1.0000 | - | - | - | 0.8770 |

A significant correlation between the content of polyphenolic compounds and antioxidant potential of the grape and that in the respective wine was observed too. The correlations between t-RVT content in the grape skins and trans-resveratrol content in the corresponding wines and between antioxidant activities of the grape skins and the corresponding wines were found to be characterized by a high correlation coefficient: 0.9291 and 0.8597, respectively (Table 4). The contribution of the stilbene to the radical scavenging capacity is crucial and was confirmed by the results obtained in this study twice. Unlike the other correlation coefficients, the correlation between TPC of wine grapes and TPC of corresponding wines was characterized by a negative coefficient (−0.8258).

Assessing the terroir influence was sufficient to compare the results obtained for wine grape samples of one and the same variety from different vineyards. Regarding the results obtained (Table 1), the varieties from Mezek, Svilengrad, surpassed the varieties from Mogilovo. For example, CSS had approximately 30% more t-RVT than CSM, but the other parameters showed similar values; SS had ca 15% more t-RVT and ca 18% more TE than SM2; MtS had ca 8% more t-RVT, ca 9% more QU and ca 15% more GAE than MtM. Differences were registered in favor of the vineyard of Mezek, Svilengrad. However, the differences were not so large. This was in accordance with differences in agro-meteorological conditions; both regions were characterized by a moderate continental climate. The advantage of Mezek, Svilengrad, was for the lower altitude and larger growing-degree-days, which seemingly contradicted to the statement, made by Bavaresco et al. [19], that t-RVT content grows with the rise of the altitude (up to 300 m a.s.l.) and by Goldberg et al. [20], with a decrease of temperature before the ripening period. The first statement was not confirmed by other researchers and the second one was rejected by the same researchers three years later [21]. On the STS, activity influences different factors and the impact of them must be considered together. The main climatic conditions differed from Pleven (North Bulgaria) to Mogilovo (South Bulgaria), where the temperature amplitude and number of the sunny days (UV radiation) occurred before the ripening period. Pleven is characterized by larger temperature amplitude and less sunny days in the veraison period.

*4.2. Content of Polyphenolic Compounds and Antioxidant Potential of Red Wines from Different Regions and in Two Consecutive Harvest Years*

Regarding the trans-resveratrol content, without considering other parameters, from one and the same vineyard in two consecutive years, Syrah was the favorite red wine: vineyard of Mogilovo, vintage 2017 and 2018—with t-RVT content 5.95 ± 0.51 and 3.86 ± 0.31 mg/L, respectively; vineyard of Pleven, vintage 2017—4.66 ± 0.4 mg/L, and vineyard Mezek, Svilengrad—3.53 ± 0.27 mg/L. The results showed that Malbec wine had a t-RVT content of 5.22 ± 0.44 mg/L (Table 2).

In many studies, Syrah showed better quality than the other tested wines. Yaman et al. [22] researched the effect of regions and determined t-RVT in red wines from different regions in Turkey, including Syrah. The found values of t-RVT ranged from 0.409 mg/L to 2.08 mg/L. The Bulgarian-tested Syrah wines showed similar and higher t-RVT content. The Syrah variety exhibited an exceptional behavior and can be considered the best candidate for the production of rich-in-stilbene wine, although some variability still occurred between the two years of the study [23].

A number of researchers compared Cabernet Sauvignon and Merlot and determined that the trans-resveratrol concentration in Merlot wines was definitely higher than in Cabernet Sauvignon [22,24,25]. The reported results ranged widely and depended on harvest year and regions. For example, t-RVT

amount in Turkish Merlot wines ranged from 0.403 to 1.52 mg/L and in Cabernet Sauvignon- from 0.319 to 0.530 mg/L [22]. In their research, Geana et al. [24] monitored trans-resveratrol in wines produced in Romania. Merlot showed 2.12 mg/L t-RVT and the lowest trans-resveratrol content was identified in Cabernet Sauvignon wine variety in the same study. In Hungarian red wines, trans-resveratrol concentration varied from 1.18 to 9.34 mg/L in Cabernet Sauvignon, while values from 2.21 to 14.32 mg/L were reported in Merlot [26]. Girelli et al. [27] determined t-RVT and QU concentrations in different Merlot wines, vintage 2009: t-RVT in Italian Merlot wines ranged from 0.43 ± 0.03 to 1.52 ± 0.09 mg/L and QU- from 1.16 ± 0.06 to 3.20 ± 0.21 mg/L. The researchers found in French Merlot'09, t-RVT in concentration of 0.49 ± 0.03 mg/L and QU—2.85 ± 0.17 mg/L; and in Rumanian Merlot'09, t-RVT and QU were not detected. Considering this statement, Bulgarian Merlot wine showed definitely higher t-RVT content than Cabernet Sauvignon wines under the same conditions, e.g., for MtMW17 and CSMW17, the determined t-RVT values were 4.65 ± 0.38 mg/L and 2.05 ± 0.18 mg/L, respectively.

From Pleven vineyard, a few selected varieties were tested, created by the methods of intra- and inter species hybridization—Rubin (RPW17), Kaylashki Rubin (KRPW17), Storgozia (SaPW17), Bouquet (BPW17), and Otel (OPW17); KRPW17 were determined t-RVT content of 4.22 ± 0.39 mg/L, very close to the t-RVT content of Syrah from the same region and vintage (SPW17)—4.66 ± 0.40 mg/L. RPW17, SaPW17 and BPW17 showed higher stilbene content than Cabernet Sauvignon—1.94 ± 0.13 mg/L (Table 2).

Quercetin values measured in red wines were reported data by different research teams: in Italian wines the values ranged from 0.03 to 0.98 mg/L [28]; in Brazilian wines—from 5.3 to 10.3mg/L [29] in Hungarian wines—from 5.80 to 13.4 mg/L [30]; in Spanish wines—from 7.4 to 9.75 mg/L [31]; and in Czech wines, QU was not detected [32]. Tsanova-Savova and Ribarova [33] reported QU contents in different Bulgarian red wines, vintage 1994–1998, including Merlot (0.7–3.2 mg/L) and Cabernet Sauvignon (2.7–3.3 mg/L). McDonald et al. [34] reported a low quercetin and a low total phenolic content of wines produced in Bulgaria. For example, in Cabernet Sauvignon, vintage 1990, found QU in the range from 0.4 to 1.2 mg/L. In the present study, the QU values were determined in range from 0.63 ± 0.05 to 3.44 ± 0.32 mg/L, and were significantly lower than this cited average concentration (Table 2).

It is interesting to take into account the influence of different factors. The main factor that determines the concentration of trans-resveratrol in the must and the wine is the balance between the level of synthesized resveratrol and degraded resveratrol concentration [14]. From the conclusion drawn in Section 4.1, the main agro-meteorological factors with a positive effect on the stilbene enrichment were a high temperature amplitude and many sunny days (UV radiation) during before-ripening-period. But despite the optimal conditions for its synthesis, the parameters of the wine making process may not be the most favorable for its storage in the final product, the wine. The main factors in the wine making with a positive effect is an extended maceration (extraction) time, and with a negative effect—the addition of clarifying agents.

To check the influence of climate condition on t-RVT and phenolic content, the results from two consecutive years were compared (Table 2). The trend was clearly visible: the vintage 2017 wines demonstrated better antioxidant properties than the vintage 2018 wines. For example, t-RVT- QU- and total phenolic contents were found in sample CSMW17 2.05 ± 0.18, 2.43 ± 0.22 and 8.17 ± 0.76 mg/L respectively, and in sample CSMW18—1.75 ± 0.15, 1.63 ± 0.12, 4.72 ± 0.41 mg/L, respectively (Table 2). The differences in the antioxidant properties in this case were due to differences in the agro-meteorological conditions for the investigated periods in the region—Village of Mogilovo, Chirpan municipality (Table 5).

**Table 5.** Agro-meteorological conditions for the investigated periods and region—Village of Mogilovo, Chirpan municipality.

| Year | Period | Temperature, °C | | | Presipitation | Rainy Days |
|---|---|---|---|---|---|---|
| | | **Min** | **Max** | **Average** | **mm** | **Number** |
| 2017 | Veraison (July) | 13.7 | 37.4 | 23.4 | 78.37 | 9 |
| | Ripening (August—September) | 7.8 | 32.5 | 22.1 | 89.62 | 5 |
| | Harvest (September—October) | 1.9 | 32.5 | 16.6 | 165.44 | 11 |
| 2018 | Veraison (July) | 22.1 | 31.9 | 22.1 | 141.25 | 18 |
| | Ripening (August—September) | 6.1 | 32.1 | 21.5 | 154.82 | 9 |
| | Harvest (September—October) | 4.8 | 32.1 | 17.9 | 47.88 | 16 |

acc. data of Branch of Space Research and Technology Institute-BAS, Stara Zagora.

The year 2017 was characterized by less rain fall and less rainy days, before ripening and during ripening period. The sunny days in moment of stilbene synthesis were more in 2017 and one of the conditions to activation of STS—UV-radiation were provided. Moreover, in the ripening period, the rainy days in 2018 caused the degradation of cumulated t-RVT. Due to the other parameters, their concentrations were probably diluted by more water content.

Comparing the results from wine samples, vintage 2017 obtained an impact of regions assessment can be made too: definitely the wine samples from Pleven, Nord Bulgaria had lower t-RVT content than the corresponding wine samples from Mogilovo, South Bulgaria (Table 2). The temperature conditions before and during the ripening period were of great significance (Table 5). The correlation between t-RVT content and temperature levels during ripening was confirmed to be inverse. The total phenolic content in the tested red wines, vintage 2017, ranged from 1.64 ± 0.14 to 10.47 ± 1.05 mmol GAE/l, and in the wines, vintage 2018—from 4.15 ± 0.39 to 5.28 ± 0.50 mmol GAE/L (Table 2).

Fernández-Pachón et al. [35] determined the antioxidant activity of wine samples by different methods. The authors assayed a total phenolic index for red wines in the range from 1262 to 2389 mg GAE/l. The main value was calculated 1877 mg GAE/l—equal to 11 mmol GAE/l. The results obtained in the present study were lower than this main value. The same authors estimated the correlation between TPC and antioxidant activity, measured by DPPH method, and found correlation coefficient 0.8237. The relationship between GAE- and TE-values obtained in the present work was found to be characterized by a similar positive coefficient—0.8399 (Table 4).

Interesting results were obtained by Ma et al. [18]: Cabernet Sauvignon wines had higher total phenolic content than Merlot wines, produced in China, but less radical scavenging capacity, determined by DPPH method. Although not quite the same in the present study, the results did confirm this statement. Comparing the results for Merlot and Cabernet Sauvignon from one harvest year and from the same producer, Cabernet Sauvignon did not always have higher TPC than Merlot wines, but the values were close. For example, harvest 2017: CSMW17 had 8.17 ± 0.76 mmol GAE/l, and MtMW17—7.55 ± 0.62 mmol GAE/l—match point for Cabernet Sauvignon. For harvest 2018: CSMW18 had 4.72 ± 0.41 mmol GAE/l, and MtMW18—5.28 ± 0.50 mmol GAE/l—match point for Merlot (Table 2). Comparing grapes and corresponding wine, the conclusion was reached: dependence up on the harvest year, the agro-meteorological conditions, and to a large extent, on the technological parameters of winemaking process.

TPC expressed as mmol GAE/L ranged in the year 2017: from 7.55 ± 0.62 to 10.47 ± 1.05 for Mogilovo, and from 1.66 ± 0.14 to 8.22 ± 0.77 for Pleven; and in 2018: from 4.15 ± 0.39 to 5.28 ± 0.50 for Mogilovo, and 4.55 ± 0.41 to 4.99 ± 0.42 for Mezek, Svilengrad (Table 2). Compared with wine grape samples, in contrary the wine samples had different trend; TPC values were similar for Merlot and Cabernet Sauvignon, but as Merlot had larger amounts, Cabernet Sauvignon, and Syrah were

not always richer than Merlot and Cabernet Sauvignon in phenolic compounds. The technological parameters of the wine making process had a stronger impact on the antioxidant concentration in wine than factors, influencing their synthesis in wine grape, like variety, agro-meteorological condition, and fungal attack. The balance between the level of synthesized and degraded and captured antioxidants was disturbed and the equilibrium was drawn to the decreasing of synthetized antioxidants and antioxidant activities.

*4.3. NIR Spectroscopy Determination of Content of Polyphenolic Compounds and Antioxidant Potential of Red Grape Varieties and Wines*

The excellent accuracy of the used NIR method for the determination of trans-resveratrol and quercetin content, and the antioxidant capacity of wine grapes and wines (Table 3, Figure 2) in the present study were demonstrated. The obtained multiple correlation coefficients R between reference and predicted on the base of NIR spectra values were bigger than 0.97 and the ratio performance deviation RPD parameter bigger than 4, which showed excellent prediction abilities of obtained equations (Table 3). NIRS models for determination of phenolic compounds in grape skins showed RPD values from 4.4 to 13.5, using fiber-optic probe and directly recording the spectra of intact grapes. There was no difference in the accuracy of determination of the investigated parameters in the samples of grape skins and wine from different grape varieties. Our results were in accordance with those reported in the literature for the determination of phenolic compounds in red wine and intact grapes and grape skins. Good calibration statistics using PLS regression models and development to quantify all polyphenol compounds from the Rias Baixas wines were reported [36,37].

To assess the accuracy of the NIR spectroscopy analysis, the standard deviation of the results from polyphenolic compounds and antioxidant potential of grapes and red wines determination by the reference methods and the SECV of NIR determination were compared. The SECV errors for all parameters have values that are in the range of SD variations. For example, SD of HPLC results for t-RVT content in red wine varied from 0.08 to 0.51 mg/L. The SECV of NIR equation for t-RVT content in red wine is 0.113 mg/L. This indicates that the accuracy of NIR spectroscopy is comparable to the accuracy of chromatographic methods. NIR spectroscopy could be a promising rapid technique in quantification of grape and wine antioxidant parameters at different stages of grape production and winemaking. After proper calibration, NIR instruments could replace the more complex chromatographic methods.

## 5. Conclusions

Today, good food criteria also include healthy capacity. In this aspect, the wine on our table should not only have good organoleptic qualities, but should be characterized by high healthy potential.

Wine has been produced in Bulgaria for more than thousand years and it is known on the international market, but data about its antioxidant potential were scarce. In the present study, three different analytical techniques were applied to analyze the content of the major polyphenols in several commercial introduced and local wine grape varieties were cultivated in different Bulgarian regions in two consecutive years. This was the first time that such an extensive experiment on Bulgarian red wine grape varieties was carried out, and a large amount of data were collected. Based on the results obtained, some conclusions were made, which would be useful for agricultural scientists, enologists and consumers:

- Bulgarian red grape varieties and wines have a high antioxidant potential;
- A positive correlation with a high regression coefficient between determined parameters: t-RVT-, QU-, and total phenolic content and antioxidant potential was found, and the contribution of t-RVT to the radical scavenging capacity was bigger than QU and TPC;
- A strong influence of agrometeorological conditions in the before-ripening-period on the accumulation of t-RVT were found;

- NIR spectroscopy could be a promising method for the antioxidants quantification of grapes and wine.

Many more wine samples must be analyzed to research the influence of the parameters of the wine making process on the preservation of antioxidants in the end product. It is known that the balance between the level of synthesized and degraded, and captured antioxidants is easy to be disturbed in the wine making process. However, how can the enologist achieve the best effect and save the balance? This can be answered in a future experiment in which the influence of each individual technological parameter on the content of antioxidants in the wine will be studied, and the NIR-technique could be applied as a rapid method for their quantification.

**Author Contributions:** Conceptualization, M.T. and V.A.; methodology M.T. and V.A.; sampling, N.G.; data collection and analysis, M.T. and S.A.; writing—original draft preparation, M.T.; writing—review and editing, V.A. and S.A. All authors have read and agreed to the published version of the manuscript.

**Funding:** This research received no external funding.

**Acknowledgments:** The authors thank Trakia University, Stara Zagora/Bulgaria (Grant No 2AF18) for funding.

**Conflicts of Interest:** The authors declare that the submitted work was carried out in the absence of any personal, professional, or financial relationships that could potentially be construed as a conflict of interest.

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
