# Peer review of "Content of Polyphenolic Compounds and Antioxidant Potential of Some Bulgarian Red Grape Varieties and Red Wines, Determined by HPLC, UV, and NIR Spectroscopy"

_agriculture, doi:10.3390/agriculture10060193_

Round 1

Reviewer 1 Report

The work concerns the characterization of the polyphenolic composition and antioxidant properties of some red grape varieties and red wines produced in three different regions of Bulgaria. The most interesting aspects concern the comparison between the antioxidant properties of grapes and wines of different cultivars and coming from different geographical areas of Bulgaria.

The work is unclear and lacking in many of its parts; a deep revision of the content and language and its resubmission to the journal are suggested.

Some suggestions for revising the text are reported below:

Title: modify the title as follows: “Antioxidant Constituents and Antioxidant Potential of Some Commercial Bulgarian Red Grape Varieties and Red Wines, Determined by HPLC, UV and NIR Spectroscopy”.

Line 34 substitute “they are passing from the grapes into the wine products” with “they pass from the solid parts of the grapes into the must”.

Line 43 substitute “quercetin (QU). It is categorized as flavonol” with “quercetin (QU) that is a flavonol”.

Lines 67-77 clarify the description of this part by highlighting which grape varieties and wines were studied and leaving the geographical origin in the background.

Line 83 better specify which winemaking processes were carry out.

Line 85. Indicate whether the wines were stored in bottle and specify the storage conditions (temperature and duration).

Line 91. Indicate which vegetal material (skins, seeds, stalks, juice) was frozen.

Line 101. Indicate the sample volume that was injected.

Line 109-111. It is recommended to report a chromatogram of the analysis.

Lines 125-133. Describe how the TE parameter was calculated and indicate the reaction time.

Lines 137-138 Indicate whether the analyses were made on fresh or defrosted skins or on the whole grape, and describe the analytical protocol.

Lines 149-157. The description of the statistical processing is not clear, please clarify it.

Paragraph 3.1. Please review the title. It is not appropriate to speak of “Antioxidant constituent”, better of “polyphenolic compounds”.

Lines 169-174. The description of the data of trans-resveratrol is confused and have to be reviewed, please add a short comment regarding the other parameters reported in Table 1.

Paragraph 3.2. Please review the title. It is not appropriate to speak of “Antioxidant constituent”, better of “polyphenolic compounds”.

Line 181. Indicate here, or in Materials and Methods, the winemaking processes that were carried on in the cellars.

Lines 179-182. A description of the data in Table 2 is missing.

Paragraph 3.3. the title is not consistent with the content of the paragraph where the calibration data of the measurements with NIR spectroscopy are reported; review it. 

Paragraph 4.1. Please review the title. It is not appropriate to speak of “Antioxidant constituent”, better of “polyphenolic compounds”. Substitute “Bulgarian Red Wine Grapes” with “Bulgarian Red Grape Varieties”.

Lines 212-213. Check the units: mg GAE/Kg dry matter or mmols/Kg FW.

Line 223. Notice: in Materials and methods FW and FM (clarify the meaning of the abbreviations) are reported, but not dm (dry matter). In any case, I suggest using DW (dry weight) instead of dm.

Lines 227-229 substitute “for the influence of t-RSV, QU and TPC on the radical” with “between t-RSV, QU and TPC and the radical”, because the presence of significant correlations does not prove the existence of a cause-effect relationship between variables.

Lines 234-235. In this line ”trans-resveratrol” is used, whereas some lines before “t-RVT” was used. Please always use the same denomination for the variables in the text.

Lines 240-242. This claim is not based on objective results.

Lines 243-259. I suppose this part is to be moved to the paragraph 4.2

Line 267. Better explain what “excellent quality” means.

Line 271. What does it mean “stilbene enriched wine”?

Line 274. Better explain what is meant by “without considering any other parameters”

Line 305-310. Indicate the agronomic and climatic factors and the winemaking conditions that affect production and loss of trans-resveratrol, specifying which have a positive and which a negative role.

Author Response

Response to reviewer #1:

Extensive editing of English language and style required

  • manuscript was checked by a professional English editing service.

Title: modify

  • It’s modified as “Content of Polyphenolic Compounds and Antioxidant Potential of Some Commercial Bulgarian Red Wine Grapes Varieties and Red Wines, Determined by HPLC, UV, and NIR Spectroscopy”

Line 34 substitute “they are passing from the grapes into the wine products”

  • It’s substituted as suggested with “they pass from the solid parts of the grapes into the must”.

Line 43 substitute “quercetin (QU). It is categorized as flavonol”

  • It’s substituted as suggested with “quercetin (QU) is a flavonol”.

Lines 67-77 clarify the description of this part

  • It’s clarified by highlighting which grape varieties and wines were studied and leaving the geographical origin in the background in section 2.1.

Line 83 better specify which winemaking processes were carry out

  • The wine making process is described by giving the main stages in section 2.1

Line 85. Indicate whether the wines were stored

  • It’s specified the bottle and the storage conditions (temperature and duration) in section 2.1

Line 91. Indicate which vegetal material (skins, seeds, stalks, juice) was frozen

  • It’s indicated in section 2.1.

Line 101. Indicate the sample volume that was injected

  • It’s indicated, 20 µl, in section 2.2.

Line 109-111. It is recommended to report a chromatogram of the analysis

  • A chromatogram is reported, Figure 1 in section 2.2.

Lines 125-133. Describe how the TE parameter was calculated and indicate the reaction time.

  • TE was calculated by regression analysis of the calibration curve. The milligrams of GAE were converted into millimoles, it’s described in section 2.3.

Lines 137-138. Indicate whether the analyses were made on fresh or defrosted skins or on the whole grape

  • The analytical protocol is described in section 2.3.

“Lines 149-157. The description of the statistical processing is not clear

  • It has been clarified in section 2.4.

Paragraph 3.1. Please review the title.

  • “Antioxidant constituent”, is substituted by “Content of polyphenolic compounds”.

Lines 169-174. The description of the data ….. in Table 1

  • All data in Table 1 are described in section 3.1.

Paragraph 3.2. Please review the title.

  • “Antioxidant constituent” is substituted by “Content of polyphenolic compounds”.

Line 181. Indicate here, or in Materials and Methods, the winemaking processes that were carried on in the cellars

  • Wine making process is described in section 2.1.

Lines 179-182. A description of the data in Table 2 is missing.

  • The data in Table 2 were described in section 3.2.

Paragraph 3.3. the title is not consistent with the content of the paragraph where the calibration data of the measurements with NIR spectroscopy are reported; review it.

  • NIR data have been reported in accordance to the paragraph content.

Paragraph 4.1. Please review the title.

  • The section title is substituted by “Content of polyphenolic compounds and antioxidant potential of red wine grapes varieties and corresponding wines”

Lines 212-213. Check the units: mg GAE/Kg dry matter or mmols/Kg FW

  • Correct is mmols/Kg FW.

Line 223. Notice: in Materials and methods FW and FM (clarify the meaning of the abbreviations) are reported, but not dm (dry matter). In any case, I suggest using DW (dry weight) instead of dm.

  • It’s technical error. It’s removed.

Lines 227-229 substitute “for the influence of t-RSV, QU and TPC on the radical” with “between t-RSV, QU and TPC and the radical”

  • It’s done.

Lines 234-235. In this line ”trans-resveratrol” is used, whereas some lines before “t-RVT” was used. Please always use the same denomination for the variables in the text

  • It’s done.

Lines 240-242. This claim is not based on objective results

  • The sentence is removed.

Lines 243-259. I suppose this part is to be moved to the paragraph 4.2

  • The data were given in Table 1 and better this part stays in section 4.1.

Line 267. Better explain what “excellent quality” means.

  • Yes, “excellent” is not the best adjective. It’s substituted by “better” and explain, that is compared to other varieties.

Line 271. What does it mean “stilbene enriched wine”?

  • Its mean “rich in stilbene”, it’s corrected.

Line 274. Better explain what is meant by “without considering any other parameters”

  • This sentence part is removed.

Line 305-310. Indicate the agronomic and climatic factors and the winemaking conditions that affect production and loss of trans-resveratrol.

  • A positive and a negative roles are specified in section 4.2.

Reviewer 2 Report

Overall, the manuscript is not well organized.

The objective of the study is not really clear and it is difficult for the reader to identify if the study is focusing on the comparison of trans-resveratrol and quercetin levels in different grape varieties or on the use of NIR spectroscopy.

The introduction and the results sections should be rewritten especially for the results where only tables have been added but no text explain those tables.

The material and methods is lacking a lot of information especially on the winemaking process which has a great influence on the concentration and type of phenolic compounds in finished wines. It is also well known that resveratrol and flavonols are at different levels depending on the grape variety and the viticultural practices.

Author Response

Thank You, for the critical revision! We have considered mostly your comments and made changes in our manuscript as you can see in our revised version. We like to share some remarks with You about this manuscript. Wine has been produced in the Bulgarian countryside for more than a thousand years. Bulgarian wine is also known on the international market but for political reasons never gained enough popularity. The amount of data of antioxidants of Bulgarian wines are scarce. We have made for the first time such an extensive research on the red wine grape varieties cultivated in different Bulgarian regions. We have used three different analytical technique to analyze the content of the major polyphenols in several introduced and local but all commercial wine grape varieties in Bulgaria. Of course, the main actor among the antioxidants is the trans-resveratrol and because of that the main focus was on it. Based on the results obtained we made some conclusions, e.g. the impact of agro-meteorological conditions, the practical application of NIR technique for quick analysis, etc. And finally, this study is a part of an extensive research project.

Round 2

Reviewer 1 Report

The authors responded to most of the previous observations, but the manuscript still needs minor revision to be suitable for publication. 

Title: The title should be shortened as follows:Antioxidant Constituents and Antioxidant Potential of Some Bulgarian Red Grape Varieties and Red Wines, Determined by HPLC, UV and NIR Spectroscopy”

Line 145. Please add the reaction time with DPPH and how the absorbance data were processed.

Paragraph 3.3. The title remains not consistent with the content of the paragraph where the results of the calibration process and not the composition of grapes or wines were reported. Please review it.

Lines 323-324. It is unclear why the Authors have removed part of the sentence, but leaving the references, without better explaining its meaning. Please review the sentence.

Author Response

Response to reviewer #1:

Title: The title should be shortened as follows: “Antioxidant Constituents and Antioxidant Potential of Some Bulgarian Red Grape Varieties and Red Wines, Determined by HPLC, UV and NIR Spectroscopy”

  • In the last review according to suggestion of the reviewer #1 we have changed title as follows: “Content of Polyphenolic Compounds and Antioxidant Potential of Some Commercial Bulgarian Red Wine Grapes Varieties and Red Wines, Determined by HPLC, UV, and NIR Spectroscopy”. Now we make it shorter: “Content of Polyphenolic Compounds and Antioxidant Potential of Some Bulgarian Red Grape Varieties and Red Wines, Determined by HPLC, UV and NIR Spectroscopy”

Line 145. Please add the reaction time with DPPH and how the absorbance data were processed.

  • The reaction time of 30 min, and absorbance measurement are added.

Paragraph 3.3. The title remains not consistent with the content of the paragraph where the results of the calibration process and not the composition of grapes or wines were reported. Please review it.

  • The title of paragraph 3.3 was changed as “Determination of content of polyphenolic compounds and antioxidant activities potential of grapes and wines by NIR Spectroscopy”

Lines 323-324. It is unclear why the Authors have removed part of the sentence, but leaving the references, without better explaining its meaning. Please review the sentence.

  • The sentence in lines 324-326 is reviewed and called: “A number of researchers compared Cabernet Sauvignon and Merlot and determined, the trans-resveratrol concentration in Merlot wines was definitely higher than in Cabernet Sauvignon [22, 24, 25].

Reviewer 2 Report

The authors made some revisions but most of the manuscript is still lacking explanation and relevance.

The experiments are described enough especially the winemaking process. The authors did not explain the yeast strain used for alcoholic fermentation. "lactic acid fermentation" does not exist and the appropriate term "malolactic fermentation" should be used.Because grape varieties grow differently and are harvested at different time, the enological parameters of each grape variety at harvest maturity including the weight of berries, the degree Brix, pH and titratable acidity is required. Because most of the results are expressed in fresh weight, the weight of berries for each variety is important. Moreover, for wine aging in barrels, it is required to specify the type of wood: oak, acacia, chestnut, etc. as well as the level of toasting. That is still not possible to compare the results obtained based on the low amount of information provided in the material and methods. 

The English requires some changes such as line 357:"the conclusion drown"; "a lot number of sunny days", etc.

Author Response

Response to reviewer #2:

“The authors did not explain the yeast strain used for alcoholic fermentation.”

  • The yeast strain used is added in line 93.

"Lactic acid fermentation" does not exist and the appropriate term "malolactic fermentation" should be used.

  • The term is corrected in line 94.

“The experiments are described enough especially the winemaking process. The authors did not explain the yeast strain used for alcoholic fermentation…………………enological parameters of each grape variety at harvest maturity including the weight of berries, the degree Brix, pH and titratable acidity is required. ……….. That is still not possible to compare the results obtained based on the low amount ……….”

  • Yes, parameters like “weight of berries, the degree Brix, pH and titratable acidity” are required by technologist to process control and to make a wine with good organoleptic parameters. But, this parameters have nothing to do with the antioxidant potential. We believe that this will shift the focus of the manuscript, the Biochemistry aspect, not the Enology. An impact of wine making process on the resveratrol content would be a new title of a new experiment. 

“The English requires some changes as line 357:"the conclusion drown"; “a lot number of sunny days"

  • Correction are made in lines 359-360: “the conclusion drawn” and “many sunny days” respectively.